# Point-of-care ultrasonography for risk stratification of non-critical suspected COVID-19 patients on admission (POCUSCO): A prospective binational study

**François Morin**[1]*, **Delphine Douillet**[1,2], **Jean François Hamel**[3,4], **Dominique Savary**[1,4], **Christophe Aubé**[5], **Karim Tazarourte**[6,7], **Kamélia Marouf**[8], **Florence Dupriez**[9], **Phillipe Le Conte**[10], **Thomas Flament**[11,12], **Thomas Delomas**[13], **Mehdi Taalba**[14], **Nicolas Marjanovic**[15,16,17], **Francis Couturaud**[18], **Nicolas Peschanski**[19], **Thomas Boishardy**[1], **Jérémie Riou**[20], **Vincent Dubée**[21,22], **Pierre-Marie Roy**[1,2]

1 Department of Emergency Medicine, University Hospital of Angers, Angers, France, 2 UNIV Angers, UMR MitoVasc CNRS 6215 INSERM 1083, Angers, France, 3 Department of Methodology and Biostatistics, University Hospital of Angers, Univ Angers, Angers, France, 4 UNIV Angers, IRSET (Institut de Recherche en Santé, Environnement et Travail) – UMR_S 1085, Angers, France, 5 Department of Radiology, University Hospital of Angers, Univ Angers, Angers, France, 6 Claude Bernard University of Lyon, Research on Healthcare Performance (RESHAPE), INSERM U1290, Lyon, France, 7 Emergency Department, Groupement Hospitalier Édouard-Herriot, Hospices Civils de Lyon, Lyon, France, 8 Department of Emergency Medicine, Hospital of Cholet, Cholet, France, 9 Department of Emergency Medicine, Cliniques Universitaires Saint-Luc, Brussels, Belgium, 10 Department of Emergency Medicine, University Hospital of Nantes, Nantes, France, 11 Department of Pulmonology University Hospital of Tours, Tours, France, 12 Société de Pneumologie de Langue Française, Chest Ultrasound Working Group (GECHO), France, 13 Department of Emergency Medicine, Hospital of Saint-Lo, Saint-Lo, France, 14 Department of Emergency Medicine, University Hospital of Rouen, Rouen, France, 15 Faculté de Médecine et de Pharmacie, Univ Poitiers, Poitiers, France, 16 Department of Emergency Medicine, University Hospital of Poitiers, Poitiers, France, 17 INSERM CIC1402 Team 5 - Acute Lung Injury and Ventilatory Support, France, 18 Department of Internal Medicine and Chest Diseases, CHU Brest, EA3878, Univ Brest, Brest, France, 19 Department of Emergency Medicine, University Hospital of Rennes, Rennes, France, 20 UNIV Angers INSERM, UMR 1066, CNRS 6021, MINT, Angers, France, 21 Infectious Diseases Department, University Hospital of Angers, Angers, France, 22 CRCINA, Univ Angers, Univ Nantes, Inserm, Nantes, France

* Francois.Morin@chu-angers.fr

**Data Availability Statement:** The datasets used and/or analyzed during the current study are available upon request, by sending your request to

## Abstract

### Background

Lung point-of-care ultrasonography (L-POCUS) is highly effective in detecting pulmonary peripheral patterns and may allow early identification of patients who are likely to develop an acute respiratory distress syndrome (ARDS). We hypothesized that L-POCUS performed within the first 48 hours of non-critical patients with suspected COVID-19 would identify those with a high-risk of worsening.

### Methods

POCUSCO was a prospective, multicenter study. Non-critical adult patients who presented to the emergency department (ED) for suspected or confirmed COVID-19 were included and had L-POCUS performed within 48 hours following ED presentation. The lung damage

the "Délégation à la Recherche Clinique et à l'Innovation Interdépartementale (DRCI)". The DRCI is a territoriality institution about clinical research, who can be requested at the email address DRCI@chu-angers.fr and who will can field data inquiries from fellow researchers.

**Funding:** This study was carried out with a grant provided by the French Ministry of Health. The sponsor had no role in the design of the study, the collection, the management, the analysis and the interpretation of the data, or the preparation of the manuscript. Apart from this grant, the authors declare no support from any organization for the submitted work. Found: (PHRC-I, April 2020, COVID19_A_001) by French Ministry of Health.

**Competing interests:** Pr. Christophe Aubé declares personal scientific collaborations with Siemens Ultrasound, outside the submitted work. Pr. Francis Couturaud declares personal consulting fees and other from BMS, personal consulting fees and other from Bayer, personal consulting fees and other from MSD, outside the submitted work. Pr. Pierre-Marie Roy declares personal fees and other from Aspen, personal fees and other from Boehringer Ingelheim, personal fees and other from Bristol Myers Squibb, other from Bayer Health Care, outside the submitted work. Other authors declare no competing interests. This does not alter our adherence to PLOS ONE policies on sharing data and materials.

severity was assessed using a previously developed score reflecting both the extension and the intensity of lung damage. The primary outcome was the rate of patients requiring intubation or who died within 14 days following inclusion.

## Results

Among 296 patients, 8 (2.7%) met the primary outcome. The area under the curve (AUC) of L-POCUS was 0.80 [95%CI:0.60–0.94]. The score values which achieved a sensibility >95% in defining low-risk patients and a specificity >95% in defining high-risk patients were <1 and ≥16, respectively. The rate of patients with an unfavorable outcome was 0/95 (0% [95%CI:0–3.9]) for low-risk patients (score = 0), 4/184 (2.17%[95%CI:0.8–5.5]) for intermediate-risk patients (score 1–15) and 4/17 (23.5%[95%CI:11.4–42.4]) for high-risk patients (score ≥16). In confirmed COVID-19 patients (n = 58), the AUC of L-POCUS was 0.97 [95% CI:0.92–1.00].

## Conclusion

L-POCUS performed within the first 48 hours following ED presentation allows risk-stratification of patients with non-severe COVID-19.

## Introduction

The COVID-19 pandemic has developed worldwide since its emergence in December 2019. Many patients have an uncomplicated course with minor symptoms, however, around 4% develop respiratory symptoms and require hospitalization [1]. Median time from illness onset to dyspnea is 6 to 8 days and around 8% of the hospitalized patients develop acute respiratory distress syndrome (ARDS), usually between Day 7 and Day 10 [2]. The rapid progression of respiratory failure soon after the onset of dyspnea is a striking feature of COVID-19 [3]. There is an urgent need for reliable tools able to early identify patients who are likely to get worse and develop ARDS.

Pulmonary computed tomography (CT-scan) appears to be very sensitive (97%) and quite specific for diagnosis of COVID-19 in patients with a clinical suspicion, provided that it is not performed within the first 4 days after symptom onset [4]. Characteristic CT-scan features are bilateral, subpleural, ground-glass opacities with air bronchograms, and ill-defined margins [5]. Those patterns can precede the positivity of the Reverse Transcriptase-Polymerase Chain Reaction (RT-PCR) for SARS-CoV-2 [6]. However, CT-scan is expensive, irradiating, requires transportation of the patient, so it cannot be used widely for early assessment of patients with COVID-19, especially in the context of hospital overcrowding.

Lung point-of-care ultrasonography (L-POCUS) is a simple, non-invasive, non-irradiating, inexpensive imaging tool available at the bedside and increasingly used by emergency physicians in their everyday clinical practice. A pre-COVID-19 study showed that L-POCUS is better than chest X-ray for detection of pneumonia and may be an alternative to the CT-scan as a screening and prognostic tool [7]. Indeed, L-POCUS is highly effective in detecting peripheral patterns and pleural abnormalities. Therefore, it could be an appropriate tool for triage of COVID-19 patients [8].

A recent publication has shown good prognostic value of lung damage estimated by L-POCUS at admission in confirmed COVID-19 population [9]. However, many patients are admitted in ED with suspected but not yet confirmed COVID-19 and need to be stratified. To

our knowledge, no robust data have been yet provided on the prognostic value of L-POCUS, in the overall population of suspected or confirmed COVID-19 patients consulting in the ED, for helping decision-making for triage in the ED [10].

The aim of this study was to determine the performance of L-POCUS at the time of ED presentation or within the first 48 hours in identifying, among patients with confirmed or highly suspected COVID-19, those who are at high-risk of adverse outcomes such as respiratory failure or death.

## Materials and methods

### Study design and settings

The point-of-care ultrasonography for risk stratification of COVID-19 patients' study (POCUSCO) was a non-interventional, prospective, multicenter study that was conducted in the ED of 11 hospitals in France and Belgium. This study was conducted in accordance with the Declaration of Helsinki. The protocol was approved by French and Belgian ethics committees, and all participants provided written informed consent. This study adheres to STROBE guidelines [11]. This study was carried out with a grant provided by the French Ministry of Health. The protocol of this study was published in the BMJ Open [12].

Patients were enrolled if they met all of the following criteria: age $\geq$ 18 years; typical COVID-19 symptoms and at least one of the three following features: i) positive SARS-CoV-2 RT-PCR, ii) typical CT-scan lesions, iii) highly-suspected COVID-19 based on the in-charge physician judgement; no requirement for respiratory support and/or other intensive care, and not subject to a limitation of care; membership of a social security scheme.

Patients for whom the follow-up at Day 14 was impossible or who had a condition making L-POCUS impossible (BMI > 35 kg/m$^2$, history of pneumonectomy) were excluded.

Patients and/or the public were not involved in the design, or conduct, or reporting or dissemination plans of this research.

### Interventions

The initial evaluation was carried out by the in-charge physician and patients were treated according to local practice. All participating patients underwent L-POCUS within the first 48 hours and a score reflecting the intensity and the extension of lung involvement was determined [13]. This score was previously developed for ARDS [13, 14]. Demographics, clinical details, and ultrasonographic findings were collected prospectively.

Patients were followed up by phone at Day 14 and their clinical status recorded according to the Ordinal Scale for Clinical Improvement for COVID-19 from the World Health Organization (WHO-OSCI) (Table 1) [15].

### Objectives and outcomes

The main objective was to assess the ability of L-POCUS to identify COVID-19 patients with a high-risk of unfavorable outcome. The primary endpoint was the development of severe COVID-19 within the 14 days after ED admission defined as a stage $\geq$ 6 on the WHO-OSCI. This stage relates to a severe inpatient requiring invasive ventilation (stage 6), and/or additional organ support (stage 7), or who died whatever the cause (stage 8). The ability of L-POCUS to predict the primary outcome occurrence was evaluated by the area under the curve (AUC) of the receiver operating characteristic (ROC) curve and its 95% confidence interval (95%CI). A sensitivity analysis was performed with the 14-day all-cause mortality rate as the outcome.

**Table 1. Ordinal Scale for Clinical Improvement (OSCI) of the World Health Organization (WHO).**

| Patient state | Descriptor | Score |
|---|---|---|
| Uninfected | No clinical or virological evidence of infection | 0 |
| Ambulatory | No limitation of activities | 1 |
| | Limitation of activities | 2 |
| Hospitalized Mild Disease | Hospitalized, no oxygen therapy | 3 |
| | Oxygen by mask or nasal prongs | 4 |
| Hospitalized Severe Disease | Non-invasive ventilation or high-flow oxygen | 5 |
| | Intubation and mechanical ventilation | 6 |
| | Ventilation + additional organ support: pressors, renal replacement therapy, ECMO... | 7 |
| Dead | Death | 8 |

The secondary objectives were: 1) To determine the threshold values of L-POCUS to stratify patients into three groups according to their risk of adverse outcome: low-risk, intermediate-risk, and high-risk patients. 2) To assess the impact of adding the result of L-POCUS evaluation to two risk-stratification clinical scores: the quick Sequential Organ Failure Assessment (qSOFA) and the CRB-65 [16, 17]. 3) To assess the impact of the knowledge and experience of the operator level (novice, confirmed or expert) on the L-POCUS performance. According to Po-Yang Tsou et al., novice sonographers were defined as physicians with no prior experiences in ultrasound and no or minimal training (≤ 7 days) in lung ultrasound; advanced ultrasonographers were defined as clinicians with more than 7 days of training in LUS, and expert ultrasonographers were defined as clinicians with a university degree in advanced lung ultrasound skills enabling them to do teaching, research and development about ultrasound [18].

We finally performed a subgroup analysis in patients for whom the diagnosis of COVID-19 was initially or subsequently confirmed by a positive RT-PCR for SARS-CoV-2.

## Lung point-of-care ultrasonography (L-POCUS)

Initial L-POCUS was performed with ultrasound scanners using low frequency (2–5 MHz) transductors. The Bedside Lung Ultrasound in an Emergency (BLUE)-Protocol was applied to patients in erect or semi-recumbent positions depending on dyspnea severity (Fig 1) [14]. Each chest wall was divided in a total of 12 areas of investigation (Fig 1). Each area was examined for at least one complete respiratory cycle. Four ultrasound aeration patterns were defined and scored 0 to 3, allowing calculation of the L-POCUS score, theoretically ranging from 0 to 36 (named Lung Ultrasound Score (LUS) by Zhao et al.) (Fig 1) [13, 19]. Considering biological risk of infection, special protective precautions were taken to protect the operator and other patients as recommended.

## Statistical analyses

Continuous variables were expressed as mean and standard deviation values. Categorical variables were described using numbers, percentages, and their 95%CI. The AUCs and their 95%CI were determined by the .632 bootstrap method. For the primary outcome, we determined in advance that the L-POCUS prognostic value would be considered as clinically relevant with a good level of evidence if the lower bound of the 95%CI of the AUC was equal to or greater than 0.7. To perform risk stratification in three groups of patients with a low, intermediate, or high-risk of an unfavorable outcome, two thresholds were calculated. The first maximized specificity with a sensitivity greater than or equal to 95% and the second maximized sensitivity with a

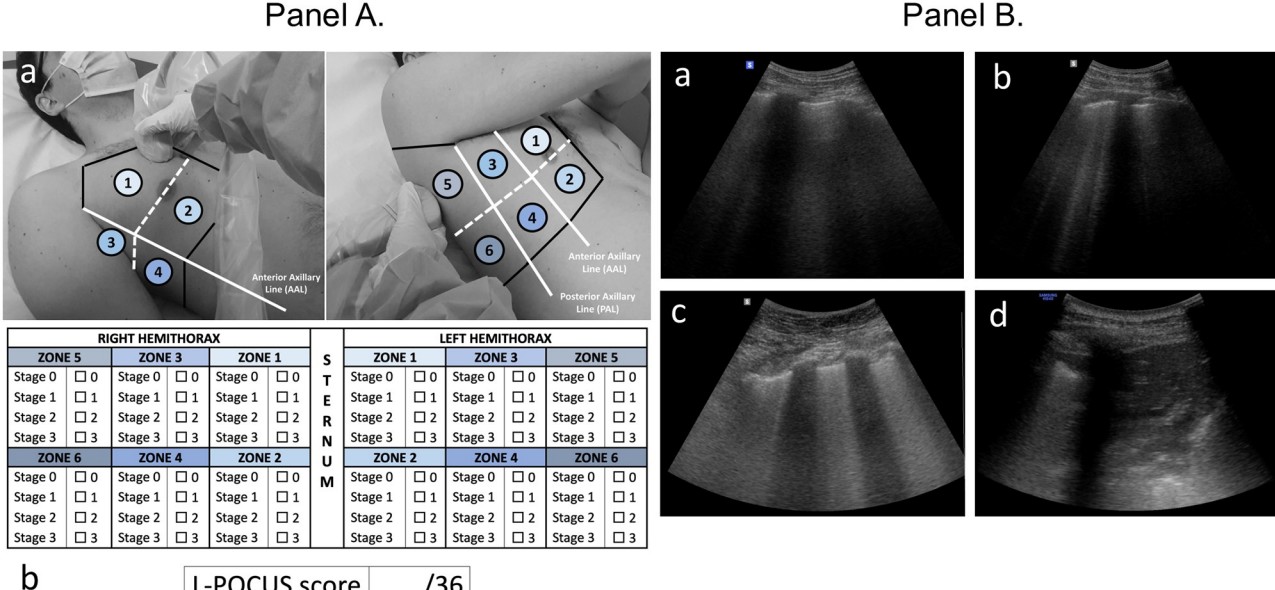

**Fig 1. Lung point-of-care ultrasonography method (L-POCUS) and examples of four ultrasound aeration stages.** (Panel A). (a). Twelve chest areas of investigation following BLUE-PLUS Protocol: *zone 1*: upper anterior chest wall; *zone 2*: lower anterior chest wall; *zone 3*: upper lateral chest wall; *zone 4*: lower lateral chest wall; *zone 5*: upper posterolateral chest wall; *zone 6*: lower posterolateral chest wall. (b) L-POCUS score grid: Each zone was examined to establish which of four ultrasound parenchymal aeration stages it exhibited, and points are assigned to them according to their severity. Stage 0 or normal aeration (0 point): Lung sliding sign associated with respiratory movement of less than 3 B lines; Stage 1 or moderate loss of lung aeration (1 point): a clear number of multiple visible B-lines with horizontal spacing between adjacent B lines ≤ 7 mm (B1 lines); Stage 2 or severe loss of lung aeration (2 points): multiple B lines fused together that were difficult to count with horizontal spacing between adjacent B lines ≤ 3 mm, including "white lung"; and Stage 3 or pulmonary consolidation (3 points): hyperechoic lung tissue, accompanied by dynamic air bronchogram. (Panel B). (a) Stage 0 or normal aeration; (b) Stage 1 or moderate loss of lung aeration; (c) Stage 2 or severe loss of lung aeration; (d) Stage 3 or pulmonary consolidation.

specificity greater than or equal to 95%. For these threshold values, sensitivity, specificity, predictive values, and likelihood ratios were assessed. To study the impact of adding the results of the L-POCUS evaluation to several risk stratification clinical rules for pulmonary infection or sepsis (qSOFA and CRB65), AUCs were compared with or without their components with a DeLong test. For this purpose, we attributed 0, 1, or 2 points in the L-POCUS result as low, moderate, or high risk according to the predefined threshold values and assessed the AUC of the risk-stratification rules with and without adding the L-POCUS result value. Assuming a rate of death or tracheal intubation requirement of 10%, and expecting an AUC of 0.8, the number of patients required to achieve a lower limit of the 95%CI, more than 0.7, was estimated as 286. Taking into consideration that 5% of patients were not followed up or could not be evaluated, the sample size was defined as 300 patients. Missing data were not imputed. A descriptive analysis of missing data was performed and compared to the available data to assess a potential bias. All statistical analyzes were performed using STATA, version 14.2; StataCorp; College Station, TX.

## Results

### Characteristics of study subjects

A total of 307 patients with suspected or confirmed COVID-19 were enrolled in this study. Among them, 2 were subsequently excluded and 9 could not be followed up (2.9%), leaving 296 patients for the main analyses (Fig 2), distributed as follows: 8.2% (24/296) with positive SARS-CoV-2 RT-PCR at admission, 5.7% (17/296) with typical CT-scan lesions and 86.1%

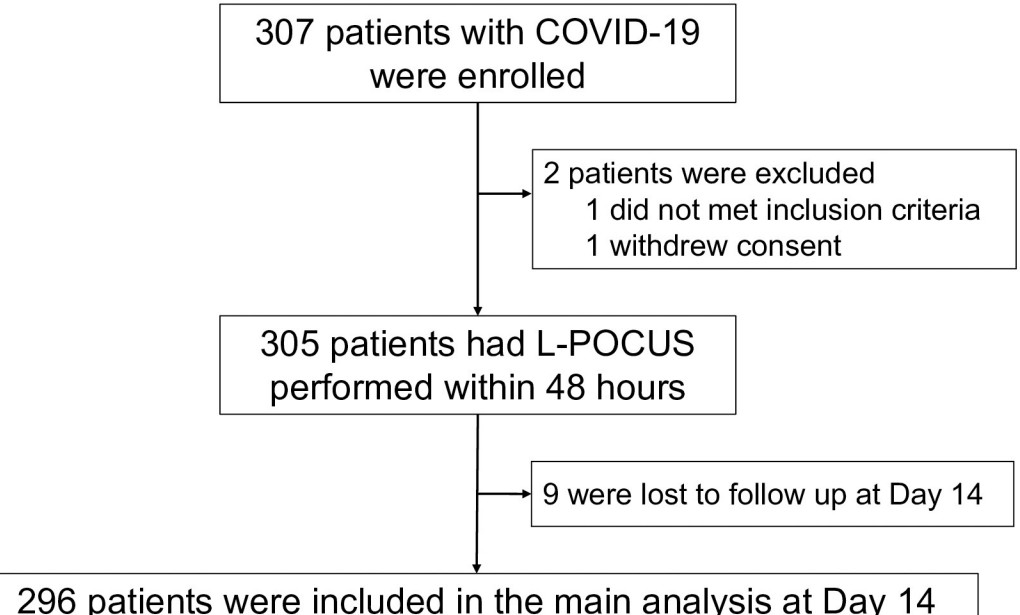

**Fig 2. Study flow chart.** COVID-19: Coronavirus disease 2019; L-POCUS: lung point of care ultrasonography; OSCI: ordinal scale for clinical improvement.

(255/296) with highly clinically suspected COVID-19 based on the in-charge physician judgement. The mean age of the overall population was 57 ± 20.8 years, and 146 (47.6%) were men (Table 2). The more common symptoms of COVID-19 were dyspnea (74.9%) cough (62.9%), abnormal thoracic auscultation (48.5%) and chest pain (40.7%).

The L-POCUS was performed by an emergency physician considered an expert, an advanced technician, and a novice in 32.2%, 44.3%, and 24.4% of cases, respectively. A CT-scan was performed in 170 patients (55.4%).

## Main results

The results of the L-POCUS are outlined in Fig 3 (Fig 3). At Day 14, among 296 analyzable patients, the main outcome had occurred in 8 patients (2.7%; seven had died, and one had required intubation and invasive ventilation). The AUC of L-POCUS was 0.80 (95%CI: 0.60–0.94) (Fig 4, Panel A). The lower value of the 95%CI did not achieve the predefined value of 0.7 necessary to consider the performance of L-POCUS as clinically relevant. In the sensitivity analysis with the 14-day all-cause mortality rate as an outcome, the AUC of L-POCUS was 0.83 (95%CI: 0.66–1). The AUC slightly increased according to the experience of the POCUS operator without significant difference: 0.86 (95%CI: 0.70–0.99), 0.82 (95%CI: 0.34–1) and 0.68 (95%CI: 0.56–0.78), for experts, confirmed or novices, respectively.

The highest L-POCUS score with a sensitivity (Se) of at least 95% was 0 point and the lowest value with a specificity (Sp) of at least 95% was 16 points. Using these cutoffs, 95 patients (32.1%) had a low-risk (score = 0) and none of them had an unfavorable outcome at Day 14 (0%[95%CI: 0.0–3.9]; Se 100%[95%CI: 63.1–100.0]; Sp 33.0%[95%CI: 27.6–38.7]; Positive likelihood ratio (LR$^+$) 1.49[95%CI: 1.4–1.6]; Negative likelihood ratio (LR$^-$) 0; Positive predictive value (PPV) 3.9%[95%CI: 3.7–4.3]; Negative predictive value (NPV) 100%). 184 patients (62.4%) had intermediate-risk (score 1 to 15) and, among them, 4 (2.17%[95%CI: 0.8–5.5]) had an unfavorable outcome (LR$^+$ 0.8[95%CI: 0.5–1.3]; LR$^-$ 1.33[95%CI: 0.8–2.1]). Finally, 17 patients (5.7%) had a high-risk (score≥16) and, among them, 4(23.5%) had an unfavorable

**Table 2. Demographic and clinical characteristics of participating patients.**

| | All patients (N = 307) |
|---|---|
| **Epidemiological characteristics** | |
| Age (years), mean ± SD | 56.94 ± 20.76 |
| Gender, N (%) | |
| Male | 146 (47.6%) |
| **Comorbidities, N (%)** | |
| Neurovascular diseases | 24 (7.8%) |
| COPD | 26 (8.5%) |
| Asthma | 46 (15.0%) |
| Hypertension | 104 (33.9%) |
| Diabetes | 37 (12.0%) |
| Active neoplasia | 18 (5.9%) |
| Chronic renal failure | 19 (6.2%) |
| Hepatic insufficiency | 6 (1.9%) |
| Chronic heart failure | 26 (8.5%) |
| **Clinical characteristics, N (%)** | |
| Confusion or GCS < 15 | 10 (3.3%) |
| Cough | 193 (62.9%) |
| Anosmia/ ageusia/ dysgeusia | 50 (16.3%) |
| Dyspnea | 230 (74.9%) |
| Rhinorrhea | 52 (16.9%) |
| Diarrhea | 61 (19.9%) |
| Abnormal pulmonary auscultation | 149 (48.5%) |
| Chest pain | 125 (40.7%) |
| **Onset of symptom to, median (IQR), days** | |
| ED admission | 8,0 (2,0–10,0) |
| **Vital parameters** | |
| Heart rate (bpm), mean ± SD | 90.67 ± 18.30 |
| SBP (mmHg), mean ± SD | 136.43 ± 22.48 |
| Temperature (°C), mean ± SD | 37.16 ± 1.00 |
| SpO2 (%), mean ± SD | 96.55 ± 3.09 |
| Respiratory rate (rpm), mean ± SD | 21.96 ± 6.08 |
| Need for supplemental oxygen, N (%) | 85 (27.7%) |

BPM: beats per minute; COPD: chronic obstructive pulmonary disease; C(U)RB-65 score: pneumonia scores based on confusion/(urea)/respiratory rate/blood pressure/age $\geq$ 65; GCS: Glasgow coma scale; IQR: interquartile range; RPM: respirations per minute; RT-PCR: reverse transcriptase polymerase chain reaction; SBP: systolic blood pressure; SD: standard deviation; SpO2: pulse-oximetry; qSOFA score: quick sepsis related organ failure assessment score.

outcome at Day 14 (23.5%[95%CI: 11.4–42.4]; Se 50%[95%CI: 15.7–84.3]; Sp 95.5%[95%CI: 92.4–97.6]; LR$^+$ 11.1[95%CI: 4.6–26.5]; LR$^-$ 0.5[95%CI: 0.3–1.1]; PPV 23.5%[95%CI: 11.4–42.4]; NPV 98.6%[95%CI: 97.2–99.3]). The proportion of patients requiring oxygen therapy was 11.6% (11/95), 33.2% (61/184) and 76.5% (13/17), in the low-risk, intermediate-risk and high-risk subgroup population, respectively.

The AUCs of the risk prediction clinical rules qSOFA and CRB65 without and with addition of the L-POCUS score were 0.52[95%CI: 0.32–0.71] and 0.75[95%CI: 0.56–0.94], and 0.72 [95%CI: 0.49–0.95] and 0.82[95%CI: 0.68–0.99], respectively.

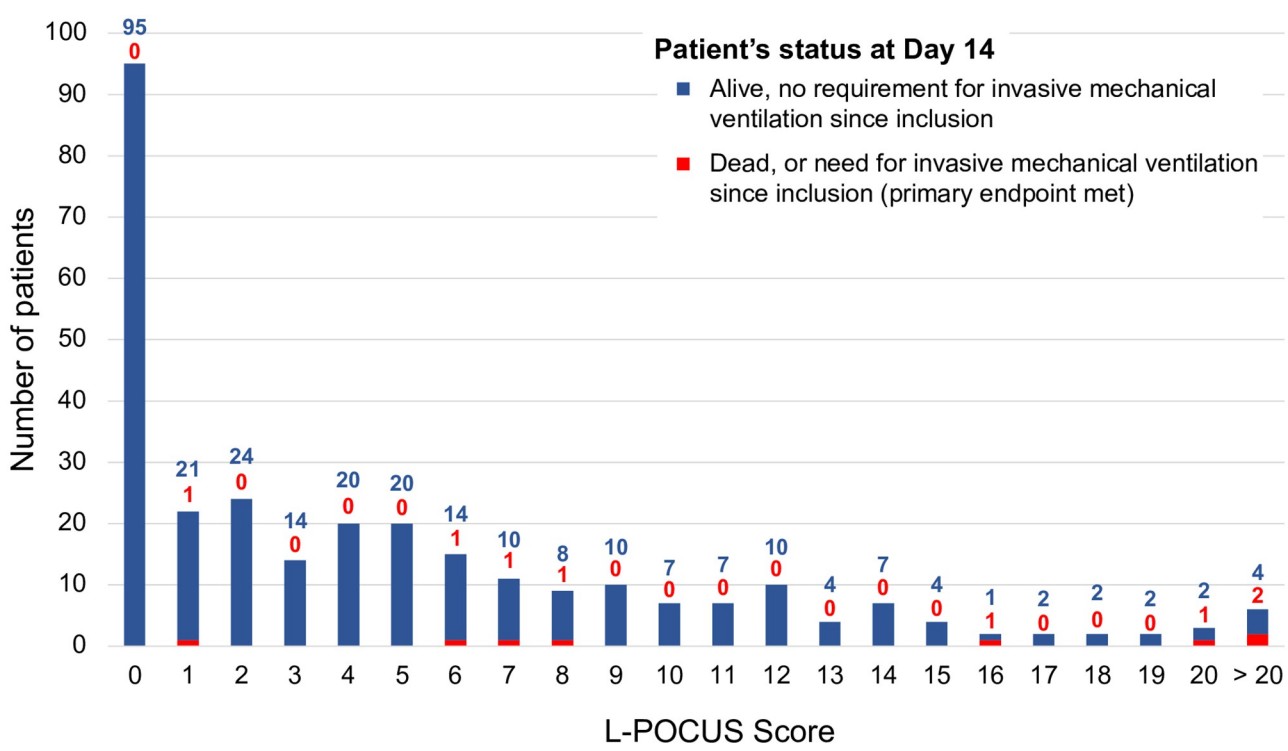

**Fig 3. Distribution of L-POCUS score according to Ordinal Scale for Clinical Improvement (OSCI) at Day 14.**

## Patients with positive SARS-CoV-2 RT-PCR

Among 240 tested patients (78.2%), 58 (24.2%) had a positive SARS-CoV-2 RT-PCR, and among them, 37.9% (22/58) needed oxygen therapy. At Day 14, 4 patients with confirmed

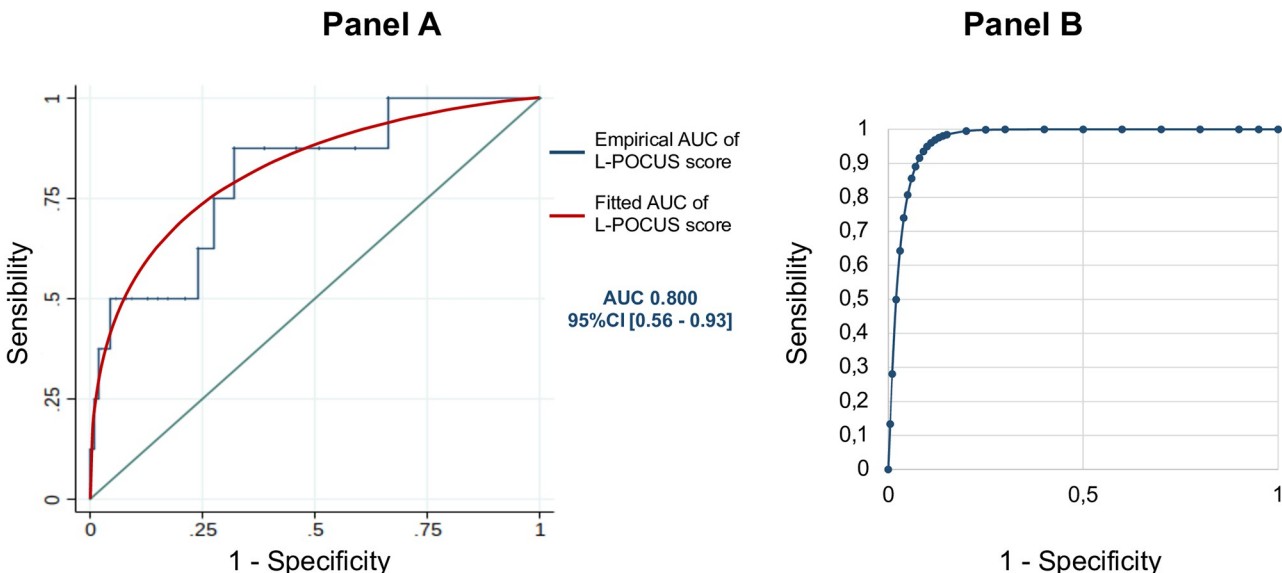

**Fig 4. L-POCUS prognostic performance.** (Panel A) Receiver operating characteristic (ROC)) curve of prognostic performance of global L-POCUS with its area-under-the-curve (AUC) and its 95% confidence interval (95%CI). (Panel B) Receiver operating characteristic (ROC) curve of prognostic performance of L-POCUS with its area-under-the-curve (AUC) and its 95% confidence interval (95%CI) for positive SARS-CoV-2 RT-PCR patients.

COVID-19 were dead (6.9%). In this population, the AUC of L-POCUS was 0.97[95%CI: 0.92–1.00] (Fig 4, Panel B). The AUC was similar in the subgroup of patients requiring oxygen therapy: 0.97[95%CI: 0.852–1.00]. Using the two thresholds defined in the overall cohort, L-POCUS determined 6 patients (10.5%) with low-risk and none of them had an unfavorable outcome at Day 14 (0%[95%CI: 0–21.5]; Se 100%[95%CI: 39.8–100]; Sp 11.3%[95%CI: 4.3–23.0]). Forty-three patients (75.4%) presented an intermediate-risk and none of them had an unfavorable outcome (0% [95%CI: 0–8.2]). Among 8 patients (14.0%) with a high-risk score, 4 had an unfavorable outcome (50.0%[95%CI: 23.7–76.3]; Se 50%[95%CI: 15.7–84.3]; Sp 92.0% [95%CI: 80.8–97.8].

## Discussion

In our prospective POCUSCO study of non-severe patients with confirmed or suspected COVID-19, L-POCUS had good results in predicting death or the need for invasive ventilation within the 14 days following ED admission and it appears to be a promising tool for risk stratification. However, because of a lower-than-expected rate of patients with an unfavorable outcome, the 95%CI of our estimates are wide, with an upper value of the AUC not achieving the predefined threshold qualifying clinical relevance with a good level of evidence.

Based on its performance in diagnosing pneumonia and ARDS, L-POCUS ought to be a useful diagnostic and risk stratification tool in the initial assessment of suspected COVID-19 patients [20, 21]. It is currently considered an alternative to physical examination for suspected COVID-19 patients in the emergency department [21]. However, this position is mainly based on expert opinion and few trials have been published. Moreover, most of them are monocentric studies assessing the correlation of L-POCUS with chest CT scans in detecting lung abnormalities suggestive of COVID-19 and/or its value in diagnosing patients with suspected COVID-19. Globally, they suggest a high sensitivity at around 90% but with a low specificity at around 25%, depending on disease prevalence [22, 23]. The integration of L-POCUS with clinical evaluation may also help to identify false-negative results occurring with RT-PCR [23]. L-POCUS would provide an rapid and effective estimate of the extent of the pulmonary histological damage [24].

To our knowledge, only one previous study assessed the performance of L-POCUS in identifying patients with confirmed COVID-19 at risk of deteriorating. Indeed, Rubio-Gracia *et al.* showed the good prognostic performances of L-POCUS to risk-stratify confirmed COVID-19 patients [9]. Our results therefore provide further important data regarding L-POCUS prognostic performances and interest for triage, especially in the overall population consulting in the ED, with confirmed COVID-19 or in a very large number of cases, in only suspected COVID-19.

Ultrasonography, including L-POCUS, was questioned for its lack of reproducibility, being dependent on the examiner. To avoid this pitfall, standardized procedures have been proposed [25]. We used a revised version of the BLUE-Protocol previously used in patients with ARDS [13]. Based on the assessment of four aeration patterns in twelve chest areas, this score is quick and easy to determine, which is particularly relevant in the ED and in the context of hospital overwhelming [13]. It is important to note that in previous studies, L-POCUS were performed by experienced emergency physicians, all certified for lung ultrasound [26]. In our trial, nearly a quarter of the exams were performed by novice physicians without any significant difference in terms of the AUC from the exams performed by experts. Indeed, a short training with 25 supervised L-POCUS helps novices acquire skills in L-POCUS [27].

With an AUC of 0.80, the global performance of L-POCUS is good in our overall population; similar results were obtained when we used death in the 14 days following inclusion as the

outcome. The prognostic performances are even better in the subgroup of patients with positive RT-PCR (lower limit of the AUC > 0.9). These results are particularly relevant in the current context of organized mass screening. When our study was performed, shortly after the start of the epidemic in Europe, PCR could be performed in a minority of the patients with suspected COVID-19. In contrast, virological confirmation of suspected COVID-19 is now rapidly available for all patients consulting in the ED. Hence, the excellent prognostic performances of L-POCUS in the population of patients with confirmed COVID-19 could be useful for initial risk-based triage in the ED. Moreover, recent data suggest that "thickening of the pleural lining, may be an important pattern for L-POCUS assessment of the prognosis of COVID-19 patients [28]. The inclusion of this criterion in a revised version of the score in a future study may improve the risk-stratification performance of L-POCUS for COVID-19 patients.

In terms of implementing L-POCUS as a triaging tool in every day clinical practice, we aimed to stratify the result into three risk categories. The rate of patients requiring oxygen therapy, at any time of their management, was proportional to the L-POCUS risk category. In comparison with low-risk category (11%), the rate of oxygen therapy was 3-fold higher in intermediate risk and 7-fold higher in high-risk category. This result strongly suggest that L-POCUS signs correspond to lung lesions. Importantly, none of the 95 patients who were determined to be low-risk (L-POCUS score = 0) suffered significant deterioration and home treatment may be suitable for these patients if they have no comorbidity or a living condition which precludes this option. It is important to note that only one patient with a L-POCUS score < 6 had an unfavorable outcome within the 14 days following ED admission. This patient was not positive with SARS-CoV-2 and died from pulmonary malignancy. On the other hand, 4 of 17 patients classified as high risk (score ≥ 16) died. All of them had a positive RT-PCR and died from COVID-19. Nevertheless, these results must be considered carefully before using L-POCUS in the early triage of COVID-19 patients, at least as a standalone tool.

In our trial the prognostic performance of the qSOFA and CRB-65 were low but the addition of the L-POCUS to these clinical rules slightly improved their performance in terms of the AUC: +0.23 for qSOFA and +0.1 for CRB-65. These results are complementary to those of Bar et al. showing that a model combining the qSOFA and ultrasound findings has good value as a diagnostic tool (AUC: 0.82 [95%CI: 0.75–0.90]) [29]. The best result was obtained with CRB-65 + L-POCUS (AUC 0.82[95%CI: 0.68–0.99]).

To our knowledge, POCUSCO is the largest multicentric, prospective study evaluating L-POCUS to risk-stratify COVID-19 patients. The most important limitation of this study is the low rate of the primary endpoint. On the basis of the first cohorts of COVID-19 inpatients, we anticipated a rate of mortality or invasive ventilation requirement of 10% [2]. It was 7% in confirmed COVID-19 patients and only 2.4% in our overall cohort. Several factors may explain this discrepancy: differences in the completeness of testing and case identification, variable thresholds for hospitalization and Intensive Care Unit admission, and improvement in patients' care [30]. Another limitation is our ne methodological choice to include patients who underwent L-POCUS within the first 48 hours of their admission. This exposed us to an unfavorable evolution of COVID-19 patients within 48 hours of their admission, before the realization of their L-POCUS. Unfortunately, we are not able to provide the proportion of L-POCUS performed at ED admission or later within the 48h. Indeed, we did not record the time of the ultrasonography. However, in the centers participating to the study, few medical wards dedicated to care of COVID-19 patients were equipped with an ultrasonography device. Therefore, it is unlikely that an important proportion of patients included in the study had their ultrasonography performed more than 24h after admission to the ED.

Still about the limitations, we excluded patients with a BMI > 35 kg/m$^2$. Yet, obesity has been identified as a condition associated with a higher risk of worsening. Moreover, only a

quarter of participating patients had a positive SARS-CoV-2 RT-PCR. The other patients may have had a minor form of COVID-19, or another less severe disease. Finally, in the absence of a derivation model, it is not methodologically justified to assess the calibration of L-POCUS [13]. Another study must be carried out to validate our results on an independent cohort.

## Conclusion

L-POCUS allows risk-stratification of suspected or confirmed COVID-19 patients. Using a 36-points score initially defined for ARDS, L-POCUS enabled identification of patients with a low-risk of deterioration (score = 0), whereas 23.6% of patients with a score $\geq$ 16 died or required invasive ventilation during the 14 days following initial evaluation. Further studies are needed to confirm these results and to determine whether a global multimodal model, integrating L-POCUS score and other considerations, would enable more accurate risk stratification of COVID-19 patients than L-POCUS score alone.

## Acknowledgments

We thank all the team of the "Maison de la Recherche Clinique" of CHU Angers and especially Sandra Merzeau and Jean-Marie Chrétien. We also thank all the research team of the Emergency Department of Angers University Hospital, and especially Cindy Augereau, Chloé Ragueneau, Clothilde Aubert and Barbara Maquin.

## Author Contributions

**Conceptualization:** François Morin, Delphine Douillet, Jean François Hamel, Vincent Dubée, Pierre-Marie Roy.

**Data curation:** François Morin.

**Formal analysis:** François Morin, Delphine Douillet, Jean François Hamel.

**Funding acquisition:** François Morin, Delphine Douillet, Jean François Hamel, Vincent Dubée.

**Investigation:** François Morin, Delphine Douillet, Jean François Hamel, Dominique Savary, Christophe Aubé, Karim Tazarourte, Kamélia Marouf, Florence Dupriez, Phillipe Le Conte, Thomas Flament, Thomas Delomas, Mehdi Taalba, Nicolas Marjanovic, Francis Couturaud, Nicolas Peschanski, Thomas Boishardy, Jérémie Riou, Vincent Dubée, Pierre-Marie Roy.

**Methodology:** François Morin, Delphine Douillet, Jean François Hamel, Dominique Savary, Jérémie Riou, Pierre-Marie Roy.

**Project administration:** François Morin.

**Resources:** François Morin.

**Supervision:** Pierre-Marie Roy.

**Validation:** François Morin, Delphine Douillet, Jean François Hamel, Dominique Savary, Jérémie Riou, Vincent Dubée, Pierre-Marie Roy.

**Visualization:** François Morin.

**Writing – original draft:** François Morin, Delphine Douillet, Jean François Hamel, Dominique Savary, Christophe Aubé, Karim Tazarourte, Kamélia Marouf, Florence Dupriez, Phillipe Le Conte, Thomas Flament, Thomas Delomas, Mehdi Taalba, Nicolas Marjanovic,

Francis Couturaud, Nicolas Peschanski, Thomas Boishardy, Jérémie Riou, Vincent Dubée, Pierre-Marie Roy.

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
