## [Decision Letter · Decision Letter 0]

4 Dec 2022

PONE-D-22-28983Point-of-care ultrasonography for risk stratification of non-critical suspected COVID-19 patients on admission (POCUSCO): a prospective binational studyPLOS ONE

Dear Dr. Morin,

Thank you for submitting your manuscript to PLOS ONE. After careful consideration, we feel that it has merit but does not fully meet PLOS ONE’s publication criteria as it currently stands. Therefore, we invite you to submit a revised version of the manuscript that addresses the points raised during the review process.

Please revise. 

We look forward to receiving your revised manuscript.

Kind regards,

Academic Editor

PLOS ONE

Journal Requirements:

"Pr. Christophe Aubé declares personal scientific collaborations with Siemens Ultrasound, outside the submitted work. Pr. Francis Couturaud declares personal consulting fees and other from BMS, personal consulting fees and other from Bayer, personal consulting fees and other from MSD, outside the submitted work. Pr. Pierre-Marie Roy declares personal fees and other from Aspen, personal fees and other from Boehringer Ingelheim, personal fees and other from 

Bristol Myers Squibb, other from Bayer Health Care, outside the submitted work.

Other authors declare no competing interests."

5. PLOS requires an ORCID iD for the corresponding author in Editorial Manager on papers submitted after December 6th, 2016. Please ensure that you have an ORCID iD and that it is validated in Editorial Manager. To do this, go to ‘Update my Information’ (in the upper left-hand corner of the main menu), and click on the Fetch/Validate link next to the ORCID field. This will take you to the ORCID site and allow you to create a new iD or authenticate a pre-existing iD in Editorial Manager. Please see the following video for instructions on linking an ORCID iD to your Editorial Manager account: https://www.youtube.com/watch?v=_xcclfuvtxQ.

Reviewers' comments:

Reviewer's Responses to Questions

**Comments to the Author**

1. Is the manuscript technically sound, and do the data support the conclusions?

Reviewer #1: Yes

Reviewer #2: Yes

2. Has the statistical analysis been performed appropriately and rigorously? 

Reviewer #1: Yes

Reviewer #2: Yes

3. Have the authors made all data underlying the findings in their manuscript fully available?

Reviewer #1: Yes

Reviewer #2: Yes

4. Is the manuscript presented in an intelligible fashion and written in standard English?

Reviewer #1: Yes

Reviewer #2: Yes

5. Review Comments to the Author

Reviewer #1: Major comments

Why did you choose the follow-up at 14 days and not until hospital discharge (or in hospital death?)?

How did you classify the emergency physician as an expert, an advanced technician or a novice? Can you provide more details about the training?

The population include mainly outpatient. Patients were not critical with only 27.7% patients who required oxygen at admission and only 2.7% met the primary outcome (death or invasive intubation). This is the main limitation of this study.

It would be interesting to explore the performance of L-POCUS only for the restricted population of COVID19 that required oxygen at admission (moderate or severe patients).

Minor comments

Page 6 line 89: What is the meaning of LUS (The lung ultrasound score ?).

Since the aim of the study is to assess L-POCUS for COVID19 patients, could you provide the proportion of the three following features: the number (%) of i) positive SARS-CoV-2 RT-

PCR, ii) typical CT-scan lesions, iii) highly-suspected COVID-19 based on the in-charge physician judgement?

Reviewer #2: The authors aimed to determine whether lung POCUS is a reliable tool for identifying COVID-19 patients at risk for worsening. To answer this question, they assessed the performance of a score based on B-line intensity (intensity of an interstitial syndrome). I guess that the authors refers to the LUS score. I think it is necessary to name it in the manuscript.

The manuscript is well written. The question is adequately introduced and the statistical analyses used seem appropriate. The outcome is robust.

Nevertheless, I have two main concerns. First of all, the delay for the realization of the ultrasound (within the first 48 hours) seems to me to bring confusion. Patient with COVID-19 can evolve unfavorably during the first 48 hours following admission to the ED. This aspect restricts the range of use of this tool, especially in ED. Could the authors specify the average time between ED admission and the completion of POCUS?

Second, using this score, the authors were able to identify patients who did not die or require mechanical ventilation within 14 days after ED admission. I have some concerns about its relevance in ED. Isn't 14 days far too long to be included in a ED-discharge decision strategy? Do the authors know the proportion of patients that required oxygen administration for each risk class (between ED admission and 14 days)? Could the authors discuss this point ?

However, despite these remarks, I think the paper will meet the criteria for publication.

6. PLOS authors have the option to publish the peer review history of their article (what does this mean?). If published, this will include your full peer review and any attached files.

Reviewer #1: **Yes: **Dr. Richard Chocron, MD, PhD

Reviewer #2: No

---

## [Author Response · Author response to Decision Letter 0]

29 Jan 2023

We thank you for your attention to our study entitled “Point-of-care ultrasonography for risk stratification of non-critical suspected COVID-19 patients on admission (POCUSCO): a prospective binational study” and for giving us the opportunity to improve our manuscript. 

Please find enclosed a point-to-point answer to the very useful comments from the Reviewers.

We think that the Reviewers suggestions and the associated revisions have substantially improved the quality of the manuscript. We hope that the editorial staff and Reviewers will find this revised version suitable for publication in the “PLOS One”.

Yours sincerely,

On behalf of all the authors, 

Dr. François Morin and Prof. Pierre-Marie Roy  

Response to reviewers’ letter

Please find below our point-by-point responses to all issues raised by the reviewers and the changes we made in our manuscript. 

Reviewer #1:

Major comments

Why did you choose the follow-up at 14 days and not until hospital discharge (or in hospital death?)?

Response to reviewers: When we designed this protocol, the scientific literature, derived from patient data infected during the first COVID-19 wave, was unclear, especially which COVID-19 patients required hospitalization. However, as only non-severe patients were included, it was foreseeable that a significant proportion of the included patients could be initially managed at home or with a very short hospitalization. Nevertheless, as observed in our trial, some of them could worsen during the following days; indeed, severe pneumonia usually occurs 7 to 14 days after the first COVID-19 symptoms. Performing a systematic follow-up at Day 14 was more constraining but allowed a rigorous and similar evaluation of our main judgement criterion for all included patients. We could also have proposed an earlier follow-up at 3 days or 7 days, because what we wanted to identify by ultrasound was a subgroup of patients with a low-risk of worsening, who could be safely managed at home. We believe that a 14-day endpoint was a good trade-off. Furthermore, the same endpoint was used in some therapeutic trials (Calvacanti et al. N Engl J Med 2020; Dubée et al. Clin Microbiol Infect 2021).

How did you classify the emergency physician as an expert, an advanced technician, or a novice? Can you provide more details about the training?

Response to reviewer: We thanks the reviewer for this question, which refers to a point that was unclear in the initial version of the manuscript. We used the definition proposed by Po-Yang Tsou et al (Po-Yang Tsou et al. Acad Emerg Med. DOI: 10.1111/acem.13818). They defined advanced ultrasonographers as clinicians with more than 7 days of training in LUS and novice sonographers as physicians with no prior experiences in ultrasound or minimal training (≤7 days) in LUS. The cutoff of 7 days of training was made by consensus among the authors based on training duration specified in the included studies. However, among the “advanced” ultrasonographers, we distinguished a group of practitioners as experts because they had a university degree in advanced lung ultrasound skills enabling them to do teaching, research, and development about ultrasound. 

The following precision has been added in “materials and methods” section of the manuscript : “According to Po-Yang Tsou et al., novice sonographers were defined as physicians with no prior experiences in ultrasound and no or minimal training (≤ 7 days) in lung ultrasound; advanced ultrasonographers were defined as clinicians with more than 7 days of training in LUS, and expert ultrasonographers were defined as clinicians with a university degree in advanced lung ultrasound skills enabling them to do teaching, research and development about ultrasound”. 

The population include mainly outpatient. Patients were not critical with only 27.7% patients who required oxygen at admission and only 2.7% met the primary outcome (death or invasive intubation). This is the main limitation of this study. It would be interesting to explore the performance of L-POCUS only for the restricted population of COVID19 that required oxygen at admission (moderate or severe patients).

Response to reviewer: We thank the reviewer for this suggestion. The AUC of L-POCUS in this subgroup of patients was 0.97 [95%CI: 0.85–1.00].

We added this result in the “Patients with Positive SARS-CoV-2 RT-PCR” section, as follows: “Among 240 tested patients (78.2%), 58 (24.2%) had a positive SARS-CoV-2 RT-PCR, and among them, 37.9% (22/58) needed oxygen therapy. At Day 14, 4 patients with confirmed COVID-19 were dead (6.9%). In this population, the AUC of L-POCUS was 0.97[95%CI: 0.92–1.00] (Fig 4, Panel B). The AUC was similar in the subgroup of patients requiring oxygen therapy: 0.97[95%CI: 0.852–1.00].”

Minor comments

Page 6 line 89: What is the meaning of LUS (The lung ultrasound score?).

Response to reviewer: We thank the reviewer for his/her vigilance and have replaced LUS by “L-POCUS” in the whole manuscript.

Since the aim of the study is to assess L-POCUS for COVID19 patients, could you provide the proportion of the three following features: the number (%) of i) positive SARS-CoV-2 RT-

PCR, ii) typical CT-scan lesions, iii) highly-suspected COVID-19 based on the in-charge physician judgement?

Response to reviewer : As proposed by the reviewer, the proportion of the three categories of included patients has been clarified in the manuscript, with the sentence: “Among them, 2 were subsequently excluded and 9 could not be followed up (2.9%), leaving 296 patients for the main analyses (Fig 2), distributed as follows: 8.2% (24/296) with positive SARS-CoV-2 RT-PCR at admission, 5.7% (17/296) with typical CT-scan lesions and 86.1% (255/296) with highly clinically suspected COVID-19 based on the in-charge physician judgement” 

Reviewer #2: 

The authors aimed to determine whether lung POCUS is a reliable tool for identifying COVID-19 patients at risk for worsening. To answer this question, they assessed the performance of a score based on B-line intensity (intensity of an interstitial syndrome). I guess that the authors refers to the LUS score. I think it is necessary to name it in the manuscript.

Response to reviewer: We agree with this comment and have added this precision in the manuscript: “Four ultrasound aeration patterns were defined and scored 0 to 3, allowing calculation of the L-POCUS score, theoretically ranging from 0 to 36 (named Lung Ultrasound Score (LUS) by Zhao et al.) (Fig 1).”

The manuscript is well written. The question is adequately introduced, and the statistical analyses used seem appropriate. The outcome is robust.

Nevertheless, I have two main concerns. 

First of all, the delay for the realization of the ultrasound (within the first 48 hours) seems to me to bring confusion. Patient with COVID-19 can evolve unfavorably during the first 48 hours following admission to the ED. This aspect restricts the range of use of this tool, especially in ED. Could the authors specify the average time between ED admission and the completion of POCUS?

Response to reviewer: We agree with this comment. The entire team, on behalf of all the inclusion centers, gives you the assurance that the vast majority of L-POCUS were carried out at the emergency department admission. Unfortunately, we are not able to provide the proportion of L-POCUS performed at ED admission or later within the 48h. Indeed, we did not record the time of the ultrasonography. However, in the centers participating to the study, few medical wards dedicated to care of COVID-19 patients were equipped with an ultrasonography device. Therefore, it is unlikely that an important proportion of patients included in the study had their ultrasonography performed more than 24h after admission to the ED. This point has been added in the limitations section of the manuscript. 

Second, using this score, the authors were able to identify patients who did not die or require mechanical ventilation within 14 days after ED admission. I have some concerns about its relevance in ED. Isn't 14 days far too long to be included in a ED-discharge decision strategy? Do the authors know the proportion of patients that required oxygen administration for each risk class (between ED admission and 14 days)? Could the authors discuss this point ?

Response to reviewer: We agree with this comment about the relevance of this tool in the context of a discharge strategy. However, when the protocol was drafted (February 2020), scientific evidence suggested that the onset time of COVID-19-related ARDS was 8–12 days, which was inconsistent with ARDS Berlin criteria, which defined a 1-week onset limit (Li et al. Crit Care. DOI : 10.1186/s13054-020-02911-9). It therefore seemed relevant to choose Day 14 because this time allowed us to know the evolution of all patients, including those who would consult within the first hours of onset of their symptoms and who would have a late adverse course. Patients usually consult in the emergency department a few days after the onset of symptoms (median delay between symptom onset and ED admission of 8 days in our study). In these patients, an unfavorable evolution in the hours or first days after admission is therefore likely and screening by pleuropulmonary ultrasound to assess this short-term risk is therefore quite interesting, especially if their RT-status COVID PCR is known (AUC 0.97[95%CI: 0.92–1.00]).

Do the authors know the proportion of patients that required oxygen administration for each risk class (between ED admission and 14 days)? Could the authors discuss this point ?

Response to reviewer: We thank the reviewer for this suggestion and we mentioned the proportion of patients requiring oxygen therapy for each risk class as follow: “ The proportion of patients requiring oxygen therapy was 11.6% (11/95), 33.2% (61/184) and 76.5% (13/17), in the low-risk, intermediate-risk and high-risk subgroup population, respectively.” A discussion of this point has been added in the discussion section: “The rate of patients requiring oxygen therapy, at any time of their management, was proportional to the L-POCUS risk category. In comparison with low-risk category (11%), the rate of oxygen therapy was 3-fold higher in intermediate-risk and 7-fold higher in high-risk L-POCUS category. This result strongly suggest that L-POCUS signs correspond to lung lesions.” In addition to this, as mentioned in the response to the first reviewer, the analyze of the performance of L-POCUS in the restricted population of positive COVID-19 patients that required oxygen at admission has been added: “Among 240 tested patients (78.2%), 58 (24.2%) had a positive SARS-CoV-2 RT-PCR, and among them, 37.9% (22/58) need oxygen therapy. At Day 14, 4 patients with confirmed COVID-19 were dead (6.9%). In this population, the AUC of L-POCUS was 0.97[95%CI: 0.92–1.00] (Fig 4, Panel B). The AUC was similar in the subgroup of patients requiring oxygen therapy: 0.97[95%CI: 0.852–1.00].”

---

## [Decision Letter · Decision Letter 1]

10 Apr 2023

Point-of-care ultrasonography for risk stratification of non-critical suspected COVID-19 patients on admission (POCUSCO): a prospective binational study

PONE-D-22-28983R1

Dear Dr. Morin,

We’re pleased to inform you that your manuscript has been judged scientifically suitable for publication and will be formally accepted for publication once it meets all outstanding technical requirements.

Kind regards,

Academic Editor

PLOS ONE

Additional Editor Comments (optional):

Reviewers' comments:

Reviewer's Responses to Questions

**Comments to the Author**

1. If the authors have adequately addressed your comments raised in a previous round of review and you feel that this manuscript is now acceptable for publication, you may indicate that here to bypass the “Comments to the Author” section, enter your conflict of interest statement in the “Confidential to Editor” section, and submit your "Accept" recommendation.

Reviewer #1: All comments have been addressed

Reviewer #2: All comments have been addressed

2. Is the manuscript technically sound, and do the data support the conclusions?

Reviewer #1: Yes

Reviewer #2: Yes

3. Has the statistical analysis been performed appropriately and rigorously? 

Reviewer #1: Yes

Reviewer #2: Yes

4. Have the authors made all data underlying the findings in their manuscript fully available?

Reviewer #1: Yes

Reviewer #2: Yes

5. Is the manuscript presented in an intelligible fashion and written in standard English?

Reviewer #1: Yes

Reviewer #2: Yes

6. Review Comments to the Author

Reviewer #1: (No Response)

Reviewer #2: The authors have responded to my suggestions and comments in a satisfactory manner. I have no further comments to make.

7. PLOS authors have the option to publish the peer review history of their article (what does this mean?). If published, this will include your full peer review and any attached files.

Reviewer #1: **Yes: **Richard CHOCRON

Reviewer #2: No

---

## [Editor Report · Acceptance letter]

14 Apr 2023

PONE-D-22-28983R1 

Point-of-care ultrasonography for risk stratification of non-critical suspected COVID-19 patients on admission (POCUSCO): a prospective binational study 

Dear Dr. Morin:

I'm pleased to inform you that your manuscript has been deemed suitable for publication in PLOS ONE. Congratulations! Your manuscript is now with our production department. 

Kind regards, 

on behalf of

Dr. Robert Jeenchen Chen 

Academic Editor

PLOS ONE